# Quantifying the Scanner-Induced Domain Gap in Mitosis Detection

**Marc Aubreville**[1]                                      MARC.AUBREVILLE@THI.DE
[1] *Technische Hochschule Ingolstadt, Ingolstadt, Germany*

**and the MIDOG challenge contributors**

## Abstract

Automated detection of mitotic figures in histopathology images has seen vast improvements, thanks to modern deep learning-based pipelines. Application of these methods, however, is in practice limited by strong variability of images between labs. This results in a domain shift of the images, which causes a performance drop of the models. Hypothesizing that the scanner device plays a decisive role in this effect, we evaluated the susceptibility of a standard mitosis detection approach to the domain shift introduced by using a different whole slide scanner. Our work is based on the MICCAI-MIDOG challenge 2021 data set, which includes 200 tumor cases of human breast cancer and four scanners.

Our work indicates that the domain shift induced not by biochemical variability but purely by the choice of acquisition device is underestimated so far. Models trained on images of the same scanner yielded an average F1 score of 0.683, while models trained on a single other scanner only yielded an average F1 score of 0.325. Training on another multi-domain mitosis dataset led to mean F1 scores of 0.52. We found this not to be reflected by domain-shifts measured as proxy A distance-derived metric.

**Keywords:** mitotic figure detection, domain shift, histopathology

## 1. Introduction

Digital histopathology images acquired at different institutions are known to feature an inherent difference in visual representation, also denoted as domain shift (Stacke et al., 2020). This variability between data sets has been associated to differences in tissue preparation or staining (Lafarge et al., 2017), and it has been identified as a major problem for model transfer and model generalization to different environments (Lafarge et al., 2017; Stacke et al., 2020). While methods counteracting this effect have recently been contributed by different working groups (Lafarge et al., 2017; Stacke et al., 2020), little attention has been paid to the disentanglement of the sources of such variability. We hypothesizes that besides staining protocols, one underestimated source of visual variability might be found in the digitization device, i.e. the microscopy whole slide image scanner. Images acquired by different scanners differ in multiple factors, including color and sharpness.

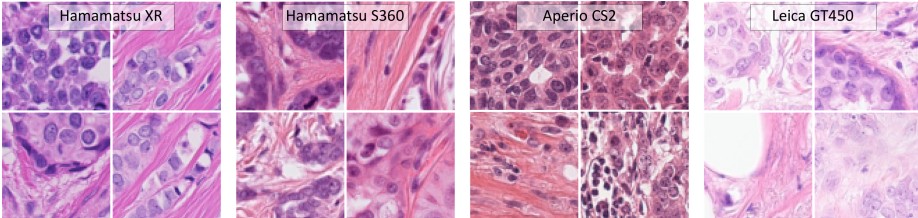

Figure 1: Random patches cropped from the different scanners of the data set.

## 2. Material and Methods

For this work, we used the publicly available MICCAI-MIDOG dataset (Aubreville et al., 2021), which includes images from 200 cases of human breast cancer, collected consecutively at a major university medical center in the Netherlands. All images were stained with standard hematoxylin and eosin stain, span an area of $2mm^2$, and were selected by a pathologist from a whole slide image, as proposed by the grading scheme for breast cancer. The images were digitized using four different scanners (50 cases per scanner). The cases from all but one scanners were, furthermore, annotated for mitotic figures by multiple experts. To quantify the domain shift and its effects on model performance, several methods have been proposed: Ben-david *et al.* considered the covariate shift of the samples by approximating the $\mathcal{H}$-divergence between the sets using a CNN classifier (Ben-David et al., 2007). This so-called proxy $\mathcal{A}$-distance (PAD) is, however, task-agnostic and the relation to performance on a specific task is strongly influenced by the learnt representations of the model, which may or may not be domain-specific. Elsahar *et al.* extended this approach to use the representations of the last hidden layer of a network that was trained on the original task and attached a classification head to discriminate the domain from those features (Elsahar and Gallé, 2019). The metric, denoted PAD*, is given as $1 - 2 \cdot \epsilon$, where $\epsilon$ is the mean absolute error on the test set. A value of 0 thus indicates a performance not exceeding guessing the domain, while values of close to 1.0 indicate a perfect differentiation of the domain, and thus a covariate shift in the features.

Mitotic figure detection is commonly regarded as an object detection task, thus we used the widespread RetinaNet approach for our investigations. We reserved 10 images of each scanner as hold-out test set. With the remaining, we trained the model and controlled our training procedure. Model selection was performed based on the Pascal VOC metric on the validation set. We repeated the training five times for each source scanner, and then ran inference with all models on the other scanners (target scanners). We calculated the F1 metric to evaluate the mitosis detection performance. Besides this actual performance measurement, we used a PAD*-derived approach for an unsupervised assessment of domain shift. For this, we used the very last hidden layer of the classification head of the network to attach a new domain classification head (two convolutional layers, a global average pooling layer, and a final fully connected layer). We trained the network until convergence (observed through a random selection of validation cases), using binary cross-entropy as loss function.

| Training set | F1 score on test set | | | PAD* score on test set | | |
|---|---|---|---|---|---|---|
| | Hamamatsu XR | Aperio CS2 | Hamamatsu S360 | Hamamatsu XR | Aperio CS2 | Hamamatsu S360 |
| TUPAC | $0.553 \pm 0.04$ | $0.613 \pm 0.05$ | $0.404 \pm 0.09$ | $0.719 \pm 0.28$ | $0.906 \pm 0.07$ | $0.945 \pm 0.09$ |
| XR | $0.578 \pm 0.03$ | $0.138 \pm 0.13$ | $0.190 \pm 0.06$ | - | $0.861 \pm 0.07$ | $0.567 \pm 0.18$ |
| CS2 | $0.390 \pm 0.10$ | $0.751 \pm 0.02$ | $0.433 \pm 0.17$ | $0.946 \pm 0.04$ | - | $0.931 \pm 0.04$ |
| S360 | $0.432 \pm 0.08$ | $0.574 \pm 0.09$ | $0.721 \pm 0.03$ | $0.769 \pm 0.06$ | $0.741 \pm 0.11$ | - |

Table 1: F1 scores and the PAD* metric of five consecutive training runs (mean $\pm$ std).

## 3. Results and Discussion

As shown in Table 1, there is a strong loss in performance, caused by the domain shift, in each of the cases. When trained on the same domain, we find a state-of-the-art recognition rate (mean F1: 0.683) using this simple object detection approach. When only a single scanner was used for training, the performance is vastly reduced when inference is performed on another scanner (mean F1: 0.359). On the other hand, training on the two-domain training set that is the TUPAC16 mitosis set improved the average out-of-domain performance significantly (mean F1: 0.523), yet still significantly below the in-domain application.

We found that the domain classifier was generally very able to discriminate domains from the last layers's features of the RetinaNet's object classifier, with a mean PAD* score of 0.820 ($\epsilon < 9\%$). However, while the models proved to be quite different, also amongst different training runs, we found the resulting PAD* metric to be unrelated to the object detection performance on the same cases (correlation coefficient: 0.039). While there is, thus, a considerable domain shift present in the features up to the last layer, it does not seem to imply deteriorated performance directly.

**Acknowledgement** The author thanks all contributors of the MIDOG challenge 2021: Christof A. Bertram, Nikolas Stathonikos, Natalie ter Hoeve, Mitko Veta, Francesco Ciompi, Robert Klopfleisch, Taryn Donovan, Christian Marzahl, Frauke Wilm, Katharina Breininger and Andreas Maier.

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
