# OpenReview forum: "Quantifying the Scanner-Induced Domain Gap in Mitosis Detection"
_MIDL.io/2021/Conference/Short — MIDL 2021 Poster_

### Official Review · Reviewer_6WxR · 2021-04-26

**Confidence:** 4
**Final Rating:** 3

**Summary:**

This paper proposes to disentangle two causes - namely staining protocols and scanner types - that are likely the cause of lack of generalization due to domain shift in histopathology. The authors empirically evaluate the impact of these domain shifts on a mitosis detection task using the MICCAI-MIDOG challenge data.


**Strengths:**

The paper is well written and the problem well motivated. This idea of disentanglement  is interesting and important to quantify. It is an important preliminary work for which I would like to see more details and results.


**Weaknesses:**

I know it is a short paper and the space is limited, but too many details are missing to be able to correctly evaluate the work/results.

The part on the domain classification is not clear. First, there is no hypothesis/motivation for this analysis and it is left to the reader to figure out why this is run. The last sentence of the paper (“While there is...”) seems to provide this hypothesis, while rejecting it.
Second, I would assume the “very last hidden layer” is a fully connected layer, not a feature mao. So how are convolution layers and GAP attached to this?
Third,the training of this model is also not clear . What parts are frozen, what hyperparameters, why convergence is evaluated on a random selection of validation cases?

The validation split is not specified.


**Deanonymize Review:**

no

**Detailed Comments:**

Other minor comments:

In introduction, it reads as if the methods (domain adversarial, color augmentation/normalization) are developed solely for staining variation. They are often referred to as staining normalization but can also work with variation due to scanner types. In Lafarge, multiple scanners are used. It doesn’t change the fact that it is interesting to disentangle the sources of shift and lack of generalization.

Is it common to refer to the metric as Pascal VOC metric? Is it mean average precision? I would refer to it as such if it is.

You could indicate the number of training and test cases in Table 1. I can’t find this information for the TUPAC dataset in the text. For the rest, iIt is already given in the text but could be good to remind it in the table.

Specify the type of correlation coefficient (Pearson?). Is it the correlation of individual pairs of domain prediction(F1)/mitosis detection(PAD)?


**Justification Of The Rating:**

The paper is interesting and relevant to the community as the problem of generalization to new data is crucial in histopathology. Important details, however, must be provided for publication as stated above.


**Paper Type:**

validation/application paper

**Special Issue:**

no

---

### Official Review · Reviewer_BqEF · 2021-05-01

**Confidence:** 3
**Final Rating:** 2

**Summary:**

The paper focuses on quantifying the impact of a scanner on the performance of a deep learning system in histopathology. The authors investigate the use of proxy A distance metric (PAD*), proposed by [1] to measure the domain shift. The metric uses the representations of the last hidden layer of a network that was trained on the original task and attached a classification head to discriminate the domain from those features.

Their findings suggest that PAD* metric is capable of identifying the presence of domain shift, however the resulting PAD* metric seems to be unrelated to the object detection performance on the same cases. Thus, even when a domain shift is present in the features up to the last layer, it does not correlate with worse performance directly.

[1]Hady Elsahar and Matthias Gall´e. To annotate or not? predicting performance drop under domain shift. In Proceedings of EMNLP-IJCNLP, pages 2163–2173, 2019.

**Strengths:**

The paper focuses on quantifying the impact of a scanner on the performance of a deep learning system in histopathology.
The paper is well written. I believe that this work could potentially be useful to the community. Metrics that quantify domain shift and correlate with performance are essential.

**Weaknesses:**

1) It is not clear how do the authors disentangle variation introduced by a scanner only from other sources of variation (staining protocol, amount of stain, its vendor, etc...). Taking into account that slides originate from different institutions I assume these factors would be difficult to control.  If it was not ensured that these other parameters where kept constant, I would not emphasize on estimating variation caused by the scanner in particular.

It would be useful to:
2) Compare the work to other methods in literature, as a baseline, like [1].
3) See the value of the PAD* metric when being tested on data from the same institution, to understand what is a base performance.
4) Mention whether data augmentation was used in training.
5) Having the code publicly available would facilitate the understanding of the paper, as well as making it more useful to the community. For now the repository with code is either private or missing.





[1]K. Stacke, G. Eilertsen, J. Unger and C. Lundström, "Measuring Domain Shift for Deep Learning in Histopathology," in IEEE Journal of Biomedical and Health Informatics, vol. 25, no. 2, pp. 325-336, Feb. 2021, doi: 10.1109/JBHI.2020.3032060.

**Deanonymize Review:**

no

**Justification Of The Rating:**

I believe that this work could potentially be useful to the community. However, while the authors attempt to quantify the impact of a scanner on domain shift in data originating from different labs, no study where the other factors causing variation were isolated was specified.

**Paper Type:**

validation/application paper

**Special Issue:**

no

---

### Meta-Review · Program_Chairs · 2021-05-11

**Recommendation:** Accept (Poster)
**Confidence:** 5

**Metareview:**

Two reviewers gave different opinions for the final rating. The authors should further supplement more experiments to prove the effectiveness of  their method (comparing the work to other methods in literature). I agree with the supporting Reviewer that this work has merits, and would maintain acceptance for this conference to raise discussion in the community.

---

### Decision · Program_Chairs · 2021-05-11

Accept (Poster)